Cross-platform normalization of microarray and RNA-seq data for machine learning applications

Thompson Jeffrey A. 1 2
Tan Jie 1 3
http://orcid.org/0000-0001-8713-9213 Greene Casey S. 1 4 5 6 csgreene@upenn.edu
1 Department of Genetics, Geisel School of Medicine at Dartmouth , Hanover, New Hampshire , United States of America
2 Quantitative Biomedical Sciences Program, Geisel School of Medicine at Dartmouth , Hanover, New Hampshire , United States of America
3 Molecular and Cellular Biology, Geisel School of Medicine at Dartmouth , Hanover, New Hampshire , United States of America
4 Department of Systems Pharmacology and Translational Therapeutics, University of Pennsylvania , Philadelphia, Pennsylvania , United States of America
5 Institute for Translational Medicine and Therapeutics, University of Pennsylvania , Philadelphia, Pennsylvania , United States of America
6 Institute for Biomedical Informatics, University of Pennsylvania , Philadelphia, Pennslyvania , United States of America
Gomez Shawn
Electronic publication date: 2016 Jan 21
Publication date: 2016
Volume: 4
Electronic Location ID: e1621
Received 2015 Oct 30; Accepted 2016 Jan 2
Copyright: © 2016 Thompson et al.
Copyright year: 2016
Copyright holder: Thompson et al.
License: This is an open access article distributed under the terms of the Creative Commons Attribution License, which permits unrestricted use, distribution, reproduction and adaptation in any medium and for any purpose provided that it is properly attributed. For attribution, the original author(s), title, publication source (PeerJ) and either DOI or URL of the article must be cited.
License URL: https://creativecommons.org/licenses/by/4.0/

Keywords: Gene expression, Normalization, RNA-sequencing, Microarray, Machine learning, Quantile normalization, Cross-platform normalization, Training, Distribution, Nonparanormal transformation

Funding: Gordon and Betty Moore Foundation’s Data-Driven Discovery Initiative GBMF4552 NIH P20 GM103534 NIH P30 CA023108 NIH UL1 TR001086 American Cancer Society Research IRG-82-003-27 This research is funded in part by the Gordon and Betty Moore Foundation’s Data-Driven Discovery Initiative through Grant GBMF4552 to CSG. JT is a Neukom Graduate Fellow supported by the William H. Neukom 1964 Institute for Computational Science. This work was supported in part by P20 GM103534, P30 CA023108 and UL1 TR001086 from the NIH and an American Cancer Society Research Grant, #IRG-82-003-27. The funders had no role in study design, data collection and analysis, decision to publish, or preparation of the manuscript.

==============================
Large, publicly available gene expression datasets are often analyzed with the aid of machine learning algorithms. Although RNA-seq is increasingly the technology of choice, a wealth of expression data already exist in the form of microarray data. If machine learning models built from legacy data can be applied to RNA-seq data, larger, more diverse training datasets can be created and validation can be performed on newly generated data. We developed Training Distribution Matching (TDM), which transforms RNA-seq data for use with models constructed from legacy platforms. We evaluated TDM, as well as quantile normalization, nonparanormal transformation, and a simple log2 transformation, on both simulated and biological datasets of gene expression. Our evaluation included both supervised and unsupervised machine learning approaches. We found that TDM exhibited consistently strong performance across settings and that quantile normalization also performed well in many circumstances. We also provide a TDM package for the R programming language.

Introduction

A wealth of gene expression data is being made publicly available by consortia such as The Cancer Genome Atlas (TCGA) (Cancer Genome Atlas Network, 2012). Such large datasets provide the opportunity to discover signals in gene expression that may not be apparent with smaller sample sizes, such as prognostic indicators or predictive factors, particularly for subsets of patients. However, discerning the signal in such large datasets frequently relies on the application of machine learning algorithms to identify relationships in high-dimensional data, or to cope with the computational complexity.

These approaches often construct a model that captures relevant features of a dataset, and the model can be used to make predictions about new data, such as how well a patient will respond to a particular treatment (Geeleher, Cox & Huang, 2014), or whether their cancer is likely to recur (Kourou et al., 2014). Therefore, the model is usually constructed using a large, diverse dataset and is then applied to incoming cases to make predictions about them.

Increasingly, investigators are measuring gene expression with RNA-seq. Despite its higher cost, several advantages of RNA-seq over DNA microarrays are typically cited (Wang, Gerstein & Snyder, 2010): RNA-seq does not require a priori knowledge of gene sequence.

RNA-seq is able to detect single nucleotide variations (Atak et al., 2013).

RNA-seq has a much higher dynamic range.

RNA-seq provides quantitative expression levels.

RNA-seq provides isoform-level expression measurements.

While RNA-seq represents a substantial technological advance, microarrays are still widely used because they are less expensive, are more consistent with historical data, and robust statistical methods exist for working with them. Perhaps more importantly, there are a tremendous number of historical microarray experiments that have already been performed. ArrayExpress, a publicly available database of experiments maintained by the European Bioinformatics Institute (EBI) (Rustici et al., 2013), contains more than 60,000 experiments and 1.8 million assays. As the transition to RNA-seq continues, the massive collection of microarray data constitute a rich resource of gene expression data. Therefore, training a classifier on large datasets created from microarrays and testing that classifier on samples measured with RNA-seq would be useful because new data could be generated with the most advanced technology and still be used for validation.

Machine learning models benefit from large, diverse training datasets in order to build generalizable models. However, most algorithms operate under the assumption that the training and test data will be drawn from the same distribution. When the distribution of training and test datasets differ, it can result in reduced fit of the model. This is referred to as dataset shift. Although some methods exist for machine learning under certain types of dataset shift (some of these are reviewed by Moreno-Torres et al. (2012)), there are no general solutions for the type of dataset shift that occurs between different gene expression platforms. In this case,(1) Ptrain(y|x)≠Ptest(y|x)∧Ptrain(x)≠Ptest(x)

where y is the class of the example and x is an expression value. This is in the category of “other types of dataset shift” mentioned by Moreno-Torres et al. (2012) for which there is no known general solution. It refers to the fact that the probability of the dependent variable may not be the same in the training and test set for a given value of an independent variable and that the probability of that value occurring is different in both datasets.

Normalization and batch correction techniques, such as quantile normalization, help to deal with some dataset shifts (Bolstad et al., 2003). Although quantile normalization was developed specifically for microarrays it has also come to be widely used for RNA-seq (Wei et al., 2014; Norton et al., 2013), as well as cross-platform normalization (Li et al., 2015; Forés-Martos et al., 2015). We also consider transformation by the nonparanormal distribution (Liu, Lafferty & Wasserman, 2009), which transforms variables using a Gaussian copula. The only methods we are aware of that were expressly designed for comparing microarrays and RNA-seq (apart from our own) are the recently published Probe Region Expression estimation Based on Sequencing (PREBS) (Uziela & Honkela, 2015) and Variance Modeling at the Observational Level (VOOM) (Law et al., 2014). PREBS estimates RNA-seq expression values at microarray probe regions in order to make the data more compatible. However, the increase in comparability means discarding the expression information contained in other reads. Additionally, because this method requires access to raw reads, it cannot be used on public data where there may be privacy concerns. Thus PREBS cannot be as widely applied to publicly available data as our method or the other methods we examine here, which only need estimated transcript abundances and do not require transcripts and probes to match. VOOM is also designed to work with raw reads, and although it is not reliant on probe regions, it is intended to be used as part of a differential expression analysis using the limma package in R (Ritchie et al., 2015). VOOM is designed to work with a priori knowledge of conditions in the data for its transformation that will often not be known in cases where clustering or classification techniques are being applied.

Given the differences in dynamic range between microarrays and RNA-seq and the fact that microarrays represent relative expression and RNA-seq quantitative counts, it may appear that the data are incommensurable. Indeed, in some cases the effect sizes for certain treatments can be dependent on the platform used (Wang et al., 2014). However, a number of papers have compared expression values from tissue samples for which both microarray and RNA-seq data have been collected. In each case, it was found that microarray and RNA-seq data are well correlated (Wang et al., 2014; Mooney et al., 2013; Malone & Oliver, 2011), although this correlation was stronger for the more highly expressed genes. Therefore, the potential for machine learning being applied cross platform should exist, given sufficient similarity in the data distributions.

Aside from other normalization techniques that might be used, microarray data are generally analyzed after log transformation, so that values represent fold-change and statistical tests requiring normality can be used. Therefore, one possible approach to integrating microarray and RNA-seq data in a machine learning pipeline would be to simply log transform the RNA-seq data. In this work, we demonstrate that this approach is insufficient to achieve consistent predictions.

In this paper, we describe Training Distribution Matching (TDM), an approach that normalizes RNA-seq data to allow models trained on microarray data to be tested on RNA-seq. We consider this approach in conjunction with three existing approaches: quantile normalization, nonparanormal transformation, and simple log transformation of the RNA-seq data. We compare performance on both simulated data and two different gene expression datasets from TCGA that contain both microarray and RNA-seq expression data. Finally, we examine how these methods perform using a model trained on a distinct microarray breast cancer dataset for which we have both microarray and RNA-seq test data.

The intuition behind TDM is to transform the RNA-seq data so that its distribution is closer to the training data but to leave between-sample relationships intact. It aims to correct the dataset shift between the microarray and RNA-seq data, using a light touch.

We evaluated all four approaches using both unsupervised and supervised machine learning methods. For an unsupervised approach we used PAM (Kaufman & Rousseeuw, 1990) and for a supervised approach we used LASSO logistic regression (Tibshirani, 1996).

Interestingly, both TDM and quantile normalization perform well, suggesting that legacy datasets may be quite useful for such analyses. However, TDM tends to hold up better than quantile normalization in cases of increased noise in the data. Nonparanormal transformation was also a strong contender, although it was designed for use specifically with the graphical lasso and may not be as applicable in the cases considered.

Methods

The basis of our approach is to adjust the distribution of RNA-seq data to improve recognition for features learned from microarrays. Most machine learning algorithms that are applicable to expression data assume the test data are drawn from the same probability distribution as the training data. If a normalization approach makes the distributions similar but does not preserve internal data dependencies, then the model will fit poorly.

Our Training Distribution Matching (TDM) approach transforms test data to have approximately the same distribution of expression values as the training data, without changing the rank order of most genes in terms of expression levels. In other words, our method is not intended to improve the rank correlation of the datasets, since this can mean changing the biological significance of the data (particularly for RNA-seq data) and brings the validity of the results into question. Instead, it is intended to improve the recognizability of features. The distribution is adjusted for the test dataset as a whole, rather than by individual sample, to avoid over-normalization.

It is to be expected that many genes will have a different rank order between datasets, regardless of the platforms used. However, by making the expression values generally more similar between datasets, the ability of a model to fit the data will be improved. Because microarray data are generally worked with as log2 transformed values, either the RNA-seq data must be log2 transformed as well, or the microarray data must not. In this work we have chosen to log2 transform the RNA-seq data, because microarray data are usually received in this form, but the package allows either decision to be made.

The Training Distribution Matching (TDM) algorithm for cross-platform normalization

TDM is a normalization method that aims to make RNA-seq data comparable with microarray data without having a large effect on inter-observation dependencies. It is performed as described in Algorithm 1.

TDM establishes a relationship between the spread of the middle half of the the training data and the extremal values, then transforms the test data to have that same relationship. It determines the ratio of the spread above the third quartile to the IQR of the training data and then uses this to bound the maximum value in the testing data (i.e. it determines the number of IQRs that can be fit between the third quartile and the maximum value). The equivalent is done for the ratio of the spread below the first quartile and the IQR of the training data, but this value is not allowed below zero. Finally, each value is mapped into a range from the minimum of the training data to the maximum of the training data (in the inverse-log space) and log2 transformed.

Quantile normalization

Quantile normalization makes it possible to ensure that two datasets are drawn from the same distribution. Given a reference distribution, a target distribution is normalized by replacing each of its values by the value of the variable with the same rank in the reference distribution. If the reference distribution contains multiple samples, the target and reference distributions will only be identical if the reference distribution is first quantile normalized across all samples.

All quantile normalization was performed using the nomalize.quantiles.use.target method of the preprocessCore package (Bolstad, 2015) in the R statistical environment (R Core Team, 2015).

The nonparanormal

The nonparanormal was designed to be used as part of an improved graphical lasso that first transforms variables to univariate smooth functions that estimate a Gaussian copula and which have been Winsorized to reduce variance (Liu, Lafferty & Wasserman, 2009). However, the transformation can be used alone for analysis and is available in the huge package for R. In essence, this method estimates a multivariate Gaussian from the data with reduced variance.

Evaluation

We evaluated the performance of our methods using both an unsupervised and a supervised machine learning algorithm. The unsupervised approach we chose was Partitioning Around Mediods (PAM). For PAM, we constructed a simulated dataset, so that we could observe the effect of TDM under controlled conditions. The supervised approach chosen for evaluation was LASSO multinomial logistic regression.

Partitioning Around Mediods (PAM)

PAM is a clustering algorithm that identifies “mediods” or examples in the dataset that represent the best centers for a user-defined number of clusters. The model that is built can be applied to new data to determine which mediod (and thus which cluster) the new data best fit to. It is similar to the k-means algorithm but tends to be more robust to outliers. Here we used the pam method from the cluster package (Maechler et al., 2015) in R.

LASSO multinomial logistic regression.

The supervised method we chose is LASSO multinomial logistic regression. For this method we relied on the glmnet package in R (Friedman, Hastie & Tibshirani, 2010). A detailed description of the method can be found in Tibshirani (1996). We evaluated the performance on classification of either tumor subtype or class depending on the dataset. In each case, we used 100 fold cross-validation to train the model. We then assessed performance of normalization methods by applying the model constructed on one platform to a dataset from a distinct platform normalized with different approaches. This process was repeated 10 times, with different random seeds.

LASSO logistic regression builds a particularly efficient model of features, using only the variables that are the most informative (Liang et al., 2013). It is a popular technique for selecting a sparse set of predictors in biological datasets (e.g. identifying the smallest set of genes that reliably predict if someone would benefit from a particular therapy). It can also be used for multinomial logistic regression, for cases in which multiple classifications are being considered. LASSO optimization is similar to normal regression, but it tends to reduce many of the coefficients of predictors to 0, leaving a relatively small set of predictors that are best able to predict an example’s class, removing redundant predictors, and leading to a model that is easier to interpret than some other approaches. This makes it particularly useful for problems like the construction of biomarkers.

Data

Simulated data

The simulated data were generated using the program SynTReN (Van den Bulcke et al., 2006). This tool enables the generation of datasets that have a distribution similar to typical microarray data, with classes in the data that are differentially expressed due to some condition and that contain correlations between gene pairs that more realistically simulate the complexity of biological data. We generated a dataset using the default settings with the following exceptions: we generated 500 genes, half of which would be background genes mostly unaffected by changing conditions; we asked for 400 samples; and we set 4 experimental conditions to be encoded in the data. Each condition received 100 samples. An additional 400 samples were created by duplicating these samples by taking the inverse log of them, rounding the results, and rescaling them to the range [0,1000000] to simulate the higher dynamic range of RNA-seq data. Although an imperfect simulation, most of what we wanted to capture is the effect of noise on datasets with matching samples but different dynamic ranges.

Additional noisy datasets were created using the addNoise method from the sdcMicro package (Templ, Kowarik & Meindl, 2015) in R on the simulated datasets by adding a percentage of gaussian noise from 0 to 3.8% in increments of .2% (i.e. 20 levels of increasing noise). The size of the increment was picked so that the correlation between each dataset and the original data approached 0 by the last increment. Each simulated RNA-seq dataset was normalized after the noise was added using TDM (with the simulated microarray data as a reference), quantile normalization (with the simulated microarray data as a target), nonparanormal transformation, or log2 transformation.

Biological data

We used three biological datasets: Dataset 1–The first contains gene expression values for tumor and tumor-adjacent normal biopsies of breast cancer from TCGA (The Cancer Genome Atlas, 2012) measured by both microarray (Agilent 244K platform) and RNA-seq (Illumina HiSeq platform). The microarray dataset contains 531 cancer samples and 63 tumor-adjacent normal samples. However, only 516 of the cancer samples and 58 of the tumor-adjacent normal samples had complete subtype data for this work, so only those were retained. The RNA-seq data included 1095 cancer samples and 113 normal. However, only 844 tumor samples and 107 normal had complete subtype data. These samples overlap 509 cancer and 60 normal samples from the microarray data. Therefore, they can be thought of as a low noise dataset for comparing results between microarray and RNA-seq. For these data, breast cancer subtype was used for classification (Cancer Genome Atlas Network, 2012).

Dataset 2–The second biological dataset contains gene expression values for tumor and tumor-adjacent normal biopsies of colon and rectal cancer from TCGA measured on the same two platforms. The microarray dataset contains 220 cancer samples and 22 normal samples, all of which were retained. The RNA-seq data included 380 cancer samples and 50 tumor-adjacent normal samples. Of these, 330 cancer samples and 29 normal included complete tumor class data and did not overlap the microarray data. In this instance, the lack of overlap was used to create a higher noise dataset. For these data CpG island methylator phenotype (CIMP) status (Sánchez-Vega et al., 2015) was used for classification.

Dataset 3–The third biological dataset is based on a breast cancer compendium created in previous work (Tan et al., 2015). It again contains both microarray and RNA-seq data. However, the first microarray dataset is from METABRIC, a retrospective cohort built from tumor banks in the UK and Canada (Curtis et al., 2012) using the Illumina HT-12 v3 platform. Missing values were imputed in these data using KNNImputer from the Sleipner library (Huttenhower et al., 2008) using 10 neighbors as recommended by Troyanskaya et al. (2001). These were filtered by median absolute deviation (MAD), keeping the 3000 genes with the highest MAD values. Of these genes, only 2520 were included in the TCGA microarray breast cancer data mentioned above, and so the METABRIC data were further filtered to include those 2520 genes. The RNA-seq data were also from the first breast cancer dataset but were filtered to include only the same 2520 genes and to include only the overlapping samples with the microarray data. This dataset allows us to compare microarray and RNA-seq data across research consortia on a set of genes selected for high variance. Furthermore, it allows us to compare the performance of normalized RNA-seq data to microarray data for the same samples.

RNA-seq and clinical data were obtained from the UCSC Cancer Browser (Goldman et al., 2013).

Results and Discussion

We developed TDM, a new method of RNA-seq data normalization intended for prediction using machine learning models built on microarray data and improved clustering. TDM performed well compared to quantile normalization, nonparanormal transformation, and log2 transformation on a range of data.

For unsupervised clustering, TDM and nonparanormal transformation are robust to noise in simulated data

TDM outperformed quantile normalization and log2 transformation on a clustering task using data simulating a matched set of 400 samples with both microarray and RNA-seq data. The data contained 4 simulated conditions and mimic the difference in dynamic range between microarrays and RNA-seq at 20 different levels of global noise (see Introduction).

Unsupervised clustering was performed using the PAM algorithm on the 400 samples with a microarray-like distribution. The accuracy of classification was assessed as the proportion of samples that were placed in a cluster in which the majority of samples matched their own class (Fig. 1). With no additional noise, all methods performed the same. However, as noise increased, the TDM transformation resulted in a more accurate clustering than quantile normalization or log2 transformation. Initially, TDM outperformed nonparanormal transformation, but as the noise continued to increase past 1.8%, nonparanormal transformation started to perform the best. Log2 transformation initially performed much worse as the noise increased, but maintained better performance at higher levels of noise.

Figure 1 The proportion of samples correctly classified in the simulated data at increasing levels of noise.

This is taken to be the proportion of samples clustered in a group for which the most common class matches their own. The x-axis represents increasing noise in the data. As the noise increases, the TDM transformed data initially have the best performance, but past 2% noise, log2 and nonparanormal transformation obtain better classification.

For additional insight into differences in clustering, we used principal coordinate analysis to visualize the similarity between samples in the data (Fig. 2). This was done at the 1.8% additional noise level (the middle noise level in Fig. 1). The figure shows that TDM resulted in slightly better separation of the 4 clusters along the first 2 principal coordinates than the other methods.

Figure 2 The two principal coordinates of simulated data at 1.8% noise level are slightly better separated into four clusters following TDM normalization than using the other methods.

(A) TDM normalization (B) log2 transformation (C) quantile normalization (D) nonparanormal transformation.

We also examined the rank correlation of genes in matching samples as the noise level increased. The mean correlation across samples is shown in Fig. S1. In general, TDM and quantile normalization maintained almost the same level of correlation as each other (compared to the training data) as the noise increased. Nonparanormal transformation and log2 transformation had lower levels of correlation as the noise increased.

Simulated data variability

Between vs. within class variability impacts the utility of data normalization methods, because if the within class variability outweighs the between class variability, it will be challenging to detect the signal of that condition in the data (Hicks & Irizarry, 2015). The distribution of expression values for each condition in the simulated data is shown with violin plots (Fig. 3), which display an appreciable level of variability both within and between classes. Quantro is a recently developed method for generating an F-score that represents the ratio of the within class variability to between class variability in the data (Hicks & Irizarry, 2015) and is available as a package for R. In particular, it provides guidance as to when quantile normalization should be applied so as to remove technical variation while minimizing the loss of biological signal. When the between class variation is low, then quantro indicates that quantile normalization should be applied. If the between class variation is much higher than the within class variation, then one must decide if it is likely to be biologically driven. If the variability is likely to be mostly technical, then quantile normalization may be effective. The simulated data provide an opportunity to assess the variability on data with tightly controlled conditions in order to better understand TDM’s performance. We ran quantro on the simulated log2 transformed RNA-seq data. The quantro score was approximately 2.01 which indicates that there is greater between class variability than within class variability and that quantile normalization may remove meaningful variability in the data if it is not mostly technical. As noise is added, the quantro score rises, eventually hitting 328.29 at 3.8% noise. This shows that as noise is added, the ratio of between class variability to within class variability rises. Of course, we know in this case that the difference in variability is technical, since we created it, but normally this information is not available, so quantro can provide useful guidance.

Figure 3 Violin plots of the simulated data distributions with standard deviation superimposed.

These plots show the distribution of expression values for the samples with each particular condition. Within and between class variability were created in the initial simulated dataset to create a challenging problem for normalization. This complication is amplified as noise is added.

Evaluation of TDM for supervised model construction using LASSO-logistic regression

For a supervised machine learning approach, we performed LASSO multinomial logistic regression to train models (on microarray datasets) for predicting tumor subtype in breast cancer and CIMP status in colon and rectal cancer, using the glmnet package (Friedman, Hastie & Tibshirani, 2010) in the R statistical environment. We then used the models to make predictions for RNA-seq datasets and the predictions were used to evaluate normalization techniques. We evaluated classification performance using the averaged values for each random seed for the total accuracy over all tumor subtypes/classes, balanced accuracy of each subtype/class (the average of the sensitivity and specificity), and Kappa statistic (classification rate after adjusting for those that could be expected by random chance).

Classification of breast cancer subtype on TCGA-only breast cancer dataset (Dataset 1)

Nonparanormal transformation resulted in the best classification performance by far on Dataset 1 (Fig. 4), with a mean total accuracy of .85 and mean Kappa of .78 for classifying samples by subtype. This was follow by TDM normalized data with a mean total accuracy of .63 and mean Kappa of .48. Third was quantile normalization with mean total accuracy of .57 and Kappa of .45. Fourth was log normalization with mean total accuracy of .54 and Kappa of .45 and finally the untransformed data with mean total accuracy of .48 and Kappa of .25.

Figure 4 Results for Dataset 1.

(A) Mean total accuracy for BRCA subtype classification across ten iterations with 95% confidence intervals. Dashed line represents the “no information rate” that could be achieved by always picking the most common class. NPN had the highest mean total accuracy on these data, followed by TDM, then quantile normalization, and log2 transformation respectively. The untransformed RNA-seq data performed the worst. (B) Mean Kappa for BRCA subtype classification across ten iterations. NPN had the highest mean Kappa on these data, followed by TDM, which was then followed by quantile normalization and log2 transformation. The untransformed RNA-seq data performed the worst.

Within each subtype, there was considerable variability as to which normalization led to the best balanced accuracy on these data (Fig. S2). Nonparanormal transformation resulted in the best classification of Basal and LumA. Quantile normalization resulted in best classification of Normal, log2 transformation resulted in best classification of Her2, and LumB (although nonparanormal transformation was about the same). It is worth nothing that the distribution of samples for each subtype varies (Fig. S3).

Classification of CIMP status on TCGA-only colon/rectal cancer dataset (Dataset 2)

TDM normalized data resulted in the highest total accuracy and Kappa on Dataset 2 (Fig. 5), although the results have wide confidence intervals. The TDM normalized data had mean total accuracy of .64, as well as mean Kappa of .36. This was very closely followed by nonparanormal transformation, and untransformed data which both had mean accuracy of .63, although they had mean Kappas of .31 and .32 respectively. Next was quantile normalization with mean total accuracy of .62, and a Kappa of just .29. Log normalization had the lowest mean total accuracy at .57, but its Kappa was the second best, at just under .36, reflecting a better diversity of classes in its results.

Figure 5 Results for Dataset 2.

(A) Mean total accuracy for colon/rectal cancer CIMP classification across ten iterations with 95% confidence intervals. Dashed line represents the “no information rate” that could be achieved by always picking the most common class. TDM had the highest mean total accuracy, although it was only slightly better than nonparanormal transformation or even the untransformed RNA-seq data. (B) TDM’s mean Kappa for colon/rectal cancer CIMP classification across ten iterations was higher than that achieved by any other method, although it was closely followed by log2 transformation.

Again, the normalization with the best balanced accuracy for specific tumor classes varied (Fig. S4). TDM resulted in best classification of CIMP (i.e. high positive CIMP status, although the untransformed data performed almost the same) and CIMPL (i.e. low positive CIMP status, although all methods were about the same). Log2 transformation had the best classification for NCIMP (i.e. non-CIMP, although TDM was close) and Normal.

Classification of breast cancer subtype training on METABRIC and testing on TCGA (Dataset 3)

TDM and quantile normalization performed almost the same on Dataset 3 (Fig. 6), with mean total accuracies of .83 and .84 respectively. These were both substantially better than the other methods, although nonparanormal transformation had mean accuracy of .78. Fourth was log2 transformation, which had a mean total accuracy of .62. Again, the untransformed data performed poorly, with a mean total accuracy of just .45, which was actually below the no information rate. Most importantly, TDM and quantile normalization were more similar in classification to a separate dataset created from microarrays on the same samples, which had a mean total accuracy of .85, than they were to any other method. Furthermore, TDM and quantile normalization both had high Kappa scores on these data at .76 and .77 respectively. These were also similar to the TCGA microarray data, with a Kappa of .78. Nonparanormal had a Kappa of .69 and log2 transformation had a Kappa of .51.

Figure 6 Results for Dataset 3 containing METABRIC microarray training data and TCGA RNA-seq test data (TDM, QN, LOG, NPN, UNTR) as well as TCGA microarray data for comparison (MA).

(A) Mean total accuracy for BRCA subtype classification across ten iterations. 95% confidence intervals shown. TDM and quantile normalization had the highest mean total accuracy for the normalized RNA-seq data when tested using a model trained on METABRIC. In fact, they were only slightly worse than actual microarray data from TCGA using the same samples. Nonparanormal transformation had the next best performance, while log2 transformation performed markedly worse. The untransformed data accuracy was actually lower than the no information rate. (B) Mean Kappa for BRCA subtype classification across ten iterations using TDM and quantile normalization achieved a high Kappa when tested using a model trained on METABRIC. They performed similarly to the TCGA microarray data (MA) that was assayed on the same samples.

For breast cancer subtypes (Fig. S5), in each case quantile normalization, TDM, and nonparanormal transformation had better balanced accuracy than log2 transformation (although it was close for Basal, LumB, and Normal). Interestingly, the untransformed data performed about the same as other methods for Her2, which was the one subtype where the TCGA microarray performed substantially better than any of the RNA-seq data.

Summary of supervised machine learning applications

TDM resulted in the best performance overall on these datasets. For Dataset 1 it was the second best performer, with the nonparanormal transformation dominating. On Dataset 2 it had the highest total accuracy and Kappa. For Dataset 3, quantile normalization had a very slightly higher total accuracy and Kappa than TDM, but only by about 1/2 of a percentage point and both were clearly better than the other methods. For analyses where an independent microarray dataset was available, cross platform (microarray to RNA-seq) performance was comparable to within platform (microarray to microarray) performance for both quantile normalization and TDM. Although the nonparanormal transformation achieved high accuracy on Dataset 1, it was only clearly superior in the unrealistic case of a dataset with samples shared between training and testing.

Discussion

RNA-seq data transformed by the TDM algorithm resulted in the most consistent performance, even though they did not end up with the highest accuracy in all cases. Nonparanormal transformation performed the best on Dataset 1 (Fig. 4), however, it is worth remembering that Dataset 1 was designed to assess how platform differences affect the results. There was substantial overlap between the actual samples in the training and test data. By transforming both the training and test data to Gaussian scores, nonparanormal transformation was best able to remove these platform differences, because the samples were largely from the same distribution after transformation. The approach did not hold up as well on the real world problems of Dataset 2 and especially Dataset 3. TDM performed the best after nonparanormal transformation on Dataset 1. On Dataset 2, TDM had the best performance, albeit with wide confidence intervals. However, TDM had the best mean Kappa on these data, revealing that this result was the less likely to result from chance. This is an important consideration on these data, given that even the untransformed dataset had close to the same accuracy but did not have as strong a Kappa score. Quantile normalization performed the best on Dataset 3, although it was almost tied with TDM for both accuracy and Kappa. Dataset 3 is important because fore this case we had both an independent microarray training set and a microarray dataset that matched the samples for the RNA-seq testing data so we could assess how RNA-seq transformation compared to testing on microarray data itself. In this case, the accuracy and Kappa of quantile normalization and TDM were clearly better than other methods and almost precisely the same as each other (with quantile normalization very slightly ahead). Additionally, these two datasets performed almost the same as the microarray dataset for the same samples. Therefore, TDM was either the top performer, or the second place method on each dataset for the real data. It also had one of the best performances on the simulated data. We considered that quantile normalization performance could be sensitive to differing distributions of classes in training and test data. However, Fig. S3 shows that the distribution is roughly the same in each for all three biological datasets. Therefore, the difference in performance is probably attributable to noise.

On the simulated data, TDM was consistently more robust against noise until the noise level hit 2%, and these results support that assessment on biological data as well. Nevertheless, nonparanormal transformation appeared to be robust to high levels of noise on the simulated data and did not perform particularly well on Dataset 2, our noisiest dataset, perhaps indicating that the levels of noise in the simulated data were eventually too high. The fact that log2 transformation also had higher accuracy than the other methods at high noise levels in the simulated data support this idea, since log2 transformation did not perform well on most tasks. It may be that log2 transformation better preserves some of the signal at high noise levels because it changes the data the least, while nonparanormal may do so by separating the marginal distributions of each gene. Overall, nonparanormal transformation and quantile normalization performed only slightly worse than TDM. In particular, if the data are filtered to remove genes with low variance before training, as with Dataset 3, our results support the use of either quantile normalization or TDM to obtain results with high accuracy. The implementation of such a step is dependent on the machine learning method used, and the goals of the study.

A factor in deciding to use quantile normalization will be the source of variance. Hicks et al. showed that when there is large variability across classes in the data and small within class variation that quantile normalization should not always be used (Hicks & Irizarry, 2015). At least some of the variance in these data may be attributable to the combination of colon and rectal cancer into a single dataset or due to difference in the distribution of subtypes and classes. In such a case, over-normalizing the data may also remove the signal. TDM provides an alternative: bring the values in the data more closely in line, while preserving inter-observation dependencies. This allows machine learning methods to better identify the signal that overcomes the noise of technical variability.

The results on Dataset 3, where both array and sequencing-based data were available, provide support for the use of the TDM algorithm for combining microarrays and RNA-seq in a single analysis. In this case, we had an additional microarray dataset measured on the same samples. TDM normalized data performed almost as well as an actual microarray dataset. This suggests that models built on data from one platform can be applied to another to generate meaningful predictions.

Conclusions

We developed TDM, a new method to normalize data so that models can be trained and evaluated without regard to platform. This will allow researchers to take advantage of the wealth of historical microarray data, including their own past experiments, as well as existing computationally derived models during the transition to next generation sequencing. We provide an R package for the transformation under the permissive open source BSD 3-clause license.

In the future, we anticipate researchers may want to apply TDM to enable analyses between RNA-seq datasets, perhaps adjusting for different quantification references. A parameter to TDM controls the granularity of the distribution matching, which should enable such analyses.

Our TDM algorithm successfully adjusts for the dataset shift that results from measurement on divergent platforms, such as that caused by the different dynamic ranges of microarrays and RNA-seq. TDM transforms the test data to have a similar distribution to the training data, while preserving most observation dependencies within those data. Because expression data are long-tailed, the compression of data near the end of the tail is expected to have a minimal impact for most machine learning methods. The consistent results with both unsupervised and supervised learning approaches on a variety of data support these conclusions and the broad utility of TDM.

Supplemental Information

Supplemental Information 1 Kendall’s tau.

Average Kendall’s tau across 20 levels of noise in the simulated data.

Click here for additional data file.

Supplemental Information 2 Dataset 1 subtype-specific accuracies.

Average balanced accuracy for BRCA subtype classification by subtype using TCGA microarray training data and TCGA RNA-seq test data.

Click here for additional data file.

Supplemental Information 3 Dataset class distributions.

The distribution of classes in the data is roughly the same between each training and testing set. (A) Distribution of classes for the first biological dataset, using TCGA microarray of breast cancer biopsies for training and TCGA RNA-seq for testing. (B) Distribution of classes for the second biological dataset, using TCGA microarray of colon/rectal cancer biopsies for training and TCGA RNA-seq for testing. (C) Distribution of classes for the third biological dataset, using METABRIC microarray of breast cancer biopsies for training and TCGA RNA-seq for testing.

Click here for additional data file.

Supplemental Information 4 Dataset 2 subtype-specific accuracies.

Average balanced accuracy for colon/rectal cancer CIMP classification using TCGA microarray for training and TCGA RNA-seq for test data.

Click here for additional data file.

Supplemental Information 5 Dataset 3 subtype-specific accuracies.

Average balanced accuracy for BRCA subtype classification by subtype using METABRIC microarray training data and TCGA RNA-seq test data as well as TCGA microarray test data for comparison.

Click here for additional data file.

Additional Information and Declarations

Competing Interests

Author Contributions

Data Deposition

The authors declare that they have no competing interests.

Jeffrey A. Thompson conceived and designed the experiments, performed the experiments, analyzed the data, wrote the paper, prepared figures and/or tables, reviewed drafts of the paper.

Jie Tan conceived and designed the experiments, contributed reagents/materials/analysis tools, reviewed drafts of the paper.

Casey S. Greene conceived and designed the experiments, analyzed the data, wrote the paper, reviewed drafts of the paper.

The following information was supplied regarding data availability:

A “TDM R Package” is provided (doi: 10.5281/zenodo.32852; url: https://github.com/greenelab/TDM).

Source code to reproduce the results of the paper called “Training Distribution Matching (TDM) Evaluation and Results” is also provided (doi: 10.5281/zenodo.32851; url: https://github.com/greenelab/TDMresults).

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
