# Peer review of "Cross-platform normalization of microarray and RNA-seq data for machine learning applications"

_PeerJ, doi:10.7717/peerj.1621_

## Round 0.1 · original submission · Minor Revisions

Please address the minor concerns and questions expressed by the reviewers.

·

Basic reporting

Author Cover Page does not specify affiliations or corresponding author.

Experimental design

No Comments

Validity of the findings

No Comments

Additional comments

This manuscript is proposes a method for rescaling RNA-Seq quantification so we can combine/improve/validate our machine learning models (e.g., network models) built on microarray-based legacy data. The proposed method, TDM, sets the upper and lower bounds on RNA-Seq result, based upon the spread (quartile distribution) of reference microarray results. This process mimics the technical limitation of microarray where the expression levels “saturate” on probes for highly expressing genes as opposed to RNA-Seq technology. TDM has a unique value, in my view, as it provides a way to prevent occasional over-normalization issue in quantile normalization approach while controlling possible outliers from simple log2-transformation.

The quality of manuscript would further improve if the authors address the following questions:

Q1: How does the clustering performance look if we also include all the mis-classified data points using Silhouette analysis after principal coordinate transformation?

Q2: How much does each noise level perturb the order with respect to Spearman correlation?

Q3: How does TDM transformation correlate to that of voom method (doi:10.1186/gb-2014-15-2-r29)?

Q4: How does the classification performance change with or without transformed RNA-Seq data (admitted that it may not be the scope of this study)?

I also suggest the authors to clarify the following:

Algorithm 1 Box: whether training data is assumed to be log2-transformed at the beginning

Line 182 and Figure 1: whether the percentage of noise was before or after each transformation

Caption of Figure 3: Add more description to “Within and between class variability is apparent”

Line 270, 279, 288, 301: The term “balanced accuracy” is used but not defined in the manuscript.

·

Basic reporting

No comments.

Experimental design

No comments.

Validity of the findings

No comments.

Additional comments

This paper addresses the question of how to make combined use of
microarray and RNAseq data for gene expression analysis, which is
particularly important given the vast amount of microarray data
that is publicly available.

The challenge is the differing distributions and characteristics of
common microarray and RNAseq data platforms, which makes straightforward
comparison impossible.

The authors make use of the fact that microarray and RNAseq data often
correlate well, and propose a fairly intuitive method they call
"Training Distribution Matching." The authors compare TDM with simple
log2 transform of the RNAseq data as well as quantile normalization,
and conclude that TDM and quantile normalization both work well but
that TDM handles noise better. The comparisons are done on a variety
of simulated and real data, comparing the results of both supervised
and unsupervised ML methods on the various transformations.

The method isn't oversold, and the comparisons seem well done. Note
that the paper is extremely reproducible (with an associated github
repo) and the implementation is freely available under a BSD license.

Unanswered questions:

Why is log2 transformation behave so erratically - any intuition?

I couldn't find any explanations as to why TDM behaved better with
respect to noisy data sets, although it seems reasonable.
Some discussion could be added here.

Bigger questions:

Could TDM be used to (for example) compare across RNAseq experiments where
different reference transcriptomes were used as quantification references?
That's likely to become a problem in the future; not everyone is going to
want to redo all of the relevant RNAseq quantification runs just to make
RNAseq directly comparable.

Minor issues:

Something odd is happening with the citations - many of them do not include
the year (e.g. Rustici et al., p 2).

---

## Round 0.2 · accepted · Accept

Thank you again for addressing the reviewers' concerns.